# K-Sort Eval: Efficient Preference Evaluation for Visual Generation via Corrected VLM-as-a-Judge

**Zhikai Li**[1,4], **Jiatong Li**[5], **Xuewen Liu**[1], **Wangbo Zhao**[4], **Pan Du**[4], **Kaicheng Zhou**[6],
**Qingyi Gu**[1,†], **Yang You**[4,†], **Zhen Dong**[3,†], **Kurt Keutzer**[2]
[1]Institute of Automation, Chinese Academy of Sciences
[2]University of California, Berkeley   [3]University of California, Santa Barbara
[4]National University of Singapore   [5]Nanyang Technological University   [6]Collov Labs

**Dataset:** https://huggingface.co/datasets/ksort/K-Sort-Eval
**Code:** https://github.com/zkkli/K-Sort-Eval

## Abstract

The rapid development of visual generative models raises the need for more scalable and human-aligned evaluation methods. While the crowdsourced Arena platforms offer human preference assessments by collecting human votes, they are costly and time-consuming, inherently limiting their scalability. Leveraging vision-language model (VLMs) as substitutes for manual judgments presents a promising solution. However, the inherent hallucinations and biases of VLMs hinder alignment with human preferences, thus compromising evaluation reliability. Additionally, the static evaluation approach lead to low efficiency. In this paper, we propose K-Sort Eval, a reliable and efficient VLM-based evaluation framework that integrates posterior correction and dynamic matching. Specifically, we curate a high-quality dataset from thousands of human votes in K-Sort Arena, with each instance containing the outputs and rankings of $K$ models. When evaluating a new model, it undergoes $(K+1)$-wise free-for-all comparisons with existing models, and the VLM provide the rankings. To enhance alignment and reliability, we propose a posterior correction method, which adaptively corrects the posterior probability in Bayesian updating based on the consistency between the VLM prediction and human supervision. Moreover, we propose a dynamic matching strategy, which balances uncertainty and diversity to maximize the expected benefit of each comparison, thus ensuring more efficient evaluation. Extensive experiments show that K-Sort Eval delivers evaluation results consistent with K-Sort Arena, typically requiring fewer than 90 model runs, demonstrating both its efficiency and reliability. The dataset and code are publicly available.

## 1 Introduction

Visual generative models have achieved remarkable progress, enabling high-quality outputs in tasks such as text-to-image (Betker et al., 2023; Podell et al., 2023; Rombach et al., 2022) and text-to-video (Esser et al., 2023; He et al., 2022; Zhou et al., 2022) generation. This rapid advancement has fueled growing interest in the field, driving the continuous emergence of new models (Zhao et al., 2025a;b; 2026) and applications (Du et al.; Li et al., 2023b; 2025a). However, the evaluation methods fail to keep pace with the model development, struggling to offer a fair and comprehensive assessment of generated outputs. Traditional metrics such as IS (Salimans et al., 2016), FID (Heusel et al., 2017), and FVD (Unterthiner et al., 2018), while widely used, are criticized for their inability to capture human preference judgements in the real world. In response, several efforts attempt to construct static datasets for human preference evaluation (Kirstain et al., 2023; Wu et al., 2023; Xu et al., 2023). However, these datasets are inherently limited by their closed-ended nature, lack of user interaction, and inability to stay up-to-date (Li et al., 2025b; Chiang et al., 2024).

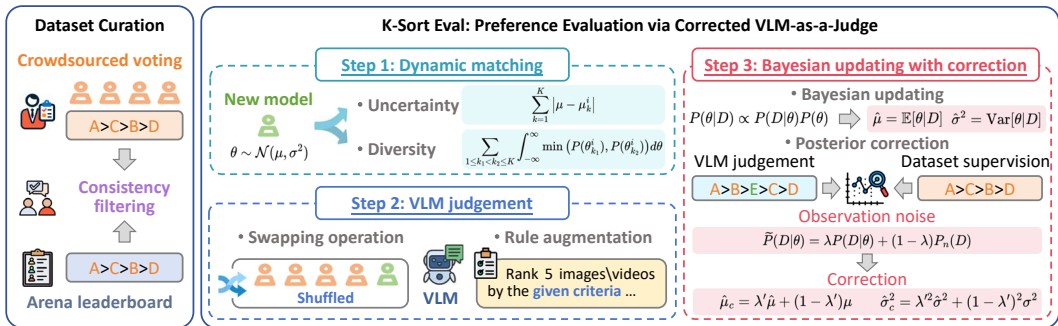

Figure 1: Overview of the proposed K-Sort Eval. First, a high-quality dataset is curated through consistency filtering. When evaluating a new model, we begin with dynamic matching to select the most informative instance. Then, two prompt strategies are employed to effectively guide the VLM and mitigate hallucinations. Finally, Bayesian updating with correction is performed, where the discrepancy between VLM prediction and dataset supervision is treated as observation noise to correct the posterior estimation of model capability.

In contrast, the Arena method, which is an open and live benchmark platform, is a more effective approach for human preference evaluation. It captures real human feedback by collecting crowd-sourced manual voting on model comparisons, allowing for a better reflection of real-world preferences. This approach is initially used for evaluating large language models (LLMs) (Chiang et al., 2024) and is later extended to apply to visual generative models (Jiang et al., 2024; Li et al., 2025b) and multi-modal models (Lu et al., 2024; Chou et al., 2024). Notably, for visual generative models, K-Sort Arena (Li et al., 2025b) significantly improves the efficiency and reliability of Arena evaluation by incorporating an improved comparison mode, probabilistic modeling and updating, and an optimized model matching strategy. Nevertheless, due to its heavy reliance on human voting, it still faces significant cost and time-consuming challenges. The excessive human involvement can also lead to potential leaderboard illusion issues (Singh et al., 2025), and the potential delays in crowdsourcing may hinder the timely evaluation of new models, thus limiting its scalability.

Therefore, leveraging powerful vision-language models (VLMs) to replace manual judgements, known as VLM-as-a-Judge (Chen et al., 2024; Liu & Zhang, 2025), presents a promising solution. For instance, T2I-CompBench (Huang et al., 2023) investigates the potential of VLMs for compositionality evaluation of text-to-image models, and VIEScore (Ku et al., 2023) demonstrates that VLMs can provide results that have a certain relevance to human evaluations. However, VLMs are inherently prone to hallucinations, inconsistencies, and biases, raising concerns about their **reliability** as a trustworthy substitute for human judgements (Li et al., 2024b; Gu et al., 2024). While certain techniques from LLM-as-a-Judge, such as swapping operation (Zheng et al., 2023), rule augmentation (Bai et al., 2022), and multi-agent collaboration (Li et al., 2023a) strategies, can be adapted, they fail to addressing these inherent issues, thus still hindering reliable alignment with human preferences. In addition, existing methods follow the static evaluation style, which necessitates processing the entire large-scale dataset, making the **efficiency** fall short of expectations.

In this paper, we propose K-Sort Eval, built on top of K-Sort Arena (Li et al., 2025b), to enable efficient and reliable visual preference evaluation via VLM-as-a-Judge. Specifically, we curate a **high-quality dataset** from thousands of human votes in K-Sort Arena, where each instance consists of the outputs of $K$ models along with their rankings. First, we use Spearman's rank correlation coefficient (Spearman, 1961) to align local rankings within each instance with the overall leaderboard, filtering out contaminated votes that significantly deviate from typical preference patterns. Then, Llama Guard (Inan et al., 2023) is applied to screen out potentially harmful or offensive user prompts, ultimately resulting in a widely applicable and representative dataset. With the above dataset, the model to be evaluated can form a $(K+1)$-wise free-for-all comparisons with the $K$ models in each instance. Notably, we propose optimized algorithms to ensure the reliability and efficiency of the evaluation. ❶ **For reliability**, we propose a posterior correction method, which utilizes the human preference outcomes in the dataset as supervision to correct the update of model capability. Following K-Sort Arena, we employ probabilistic modeling to represent model capabilities, and use VLM judgements as observations to update the posterior probability via Bayesian

Table 1: Comparison with existing evaluation methods. The proposed K-Sort Eval demonstrates advantages in terms of scalability, alignment, efficiency, and generalizability.

| Evaluation Method | Pipeline (Scalability) | Judgement (Alignment) | Data Selection (Efficiency) | Target Model (Generalizability) |
|---|---|---|---|---|
| K-Sort Arena (Li et al., 2025b) | Manual | Human | No dataset | Image & Video |
| TIFA (Hu et al., 2023), T2I-CompBench (Huang et al., 2023) | Automatic | Predefined metric | Static | Image |
| VBench (Huang et al., 2024), EvalCrafter (Liu et al., 2024) | Automatic | Predefined metric | Static | Video |
| ImageReward (Xu et al., 2023), HPD (Wu et al., 2023) | Automatic | Reward model | Static | Image |
| GenAI-Bench (Li et al., 2024a) | Automatic | Reward model | Static | Image & Video |
| VIEScore (Ku et al., 2023), MiniGPT4-CoT (Huang et al., 2023) | Automatic | VLM judge | Static | Image |
| VideoPhy (Bansal et al., 2024), VideoScore (He et al., 2024) | Automatic | VLM judge | Static | Video |
| K-Sort Eval (Ours) | Automatic | Corrected VLM judge | Dynamic | Image & Video |

inference. Here, we treat the misalignment between VLM results and supervision as observation noise, and thus derive an adaptive correction policy for the posterior probability. ❷ **For efficiency**, we propose a dynamic matching strategy, which leverages both uncertainty and diversity to promote the comparison of maximum expected gains. It avoids worthless comparisons, enabling the evaluation to be accomplished using only a subset of the dataset, without traversing the entire dataset. The overview of K-Sort Eval is illustrated in Figure 1.

Table 1 compares K-Sort Eval with existing evaluation methods across various categories, highlighting its advantages in scalability, alignment, efficiency, and generalizability. Furthermore, we conduct extensive experiments to validate the effectiveness of K-Sort Eval, and the results show that it achieves results consistent with K-Sort Arena, while requiring fewer than 90 model runs in most cases, demonstrating its strong potential for reliable preference evaluation of generative models.

## 2 RELATED WORK

**Visual Generation Evaluation.** Traditional metrics assess the quality of generated content by measuring its divergence from real data, with FID (Heusel et al., 2017) and IS (Salimans et al., 2016) commonly used for images, and FVD (Unterthiner et al., 2018) for videos. To enable more comprehensive evaluations, various benchmarks have been proposed, including image benchmarks such as TIFA (Hu et al., 2023) and T2I-CompBench (Huang et al., 2023), as well as video benchmarks like VBench (Huang et al., 2024) and EvalCrafter (Liu et al., 2024). However, these benchmarks still rely on predefined metrics, which typically fail to reflect human preferences. Several efforts focus on developing reward models, such as ImageReward (Xu et al., 2023), HPD (Wu et al., 2023), Pick-a-Pic (Kirstain et al., 2023), and GenAI-Bench (Li et al., 2024a), which finetune the CLIP model (Radford et al., 2021) to achieve better alignment. However, CLIP's limited ability to capture high-level semantics continues to hinder alignment and fairness in evaluation.

**Arena Evaluation with Human Preferences.** To enable evaluations that better align with human preferences, Chatbot Arena (Chiang et al., 2024) builds a platform for anonymized pairwise comparisons of language models, and collects user judgements on the outputs to obtain an overall model ranking. This approach also inspires efforts in other domains, such as WildVision (Lu et al., 2024) for multi-modal models and GenAI Arena (Jiang et al., 2024) for visual generative models. Furthermore, K-Sort Arena (Li et al., 2025b) introduces $K$-wise comparisons ($K > 2$), leveraging probabilistic modeling and matching strategies to enable more efficient and reliable evaluation of visual generative models. Despite their success, these methods are resource-intensive and time-consuming, leading to evaluation delays and potential issues such as leaderboard overfitting or illusion (Singh et al., 2025), which inherently limit their scalability.

**Large Model as a Judge.** In addition to generation, the judgement capabilities of LLMs, called LLM-as-a-judge, have also been explored for scoring and ranking tasks (Li et al., 2024b). To address the hallucinations and biases issues, various strategies have been developed, including swapping operations (Zheng et al., 2023), rule augmentation (Bai et al., 2022), multi-agent collaboration (Li et al., 2023a), demonstrations (Jain et al., 2023), and multi-turn interactions (Bai et al., 2023b). Likewise, leveraging VLMs as judge models to harness their visual understanding capabilities has shown great promise (Chen et al., 2024; Liu & Zhang, 2025). VLMs have been employed to evaluate the quality of generated images (Ku et al., 2023; Huang et al., 2023) and videos (Bansal et al., 2024; He et al., 2024). However, VLMs still exhibit inherent hallucinations and biases, which limit their

ability to make judgments fully aligned with human preferences. Thus, how to utilize VLMs for reliable human preference evaluations remains an open issue.

## 3 METHODOLOGY

In this paper, we propose K-Sort Eval, an efficient VLM-as-a-Judge evaluation framework that reliably aligns human preferences. K-Sort Eval benefits from both new datasets and novel evaluation strategies. The dataset curation is presented in Section 3.1, followed by the proposed methods for improving evaluation reliability and efficiency in Sections 3.2 and 3.3, respectively, with the overall evaluation pipeline ultimately formed in Section 3.4.

### 3.1 HUMAN PREFERENCE DATASET CURATION

K-Sort Arena (Li et al., 2025b), as a precursor to this work, serves as the platform for collecting data on human preferences. K-Sort Arena organizes free-for-all comparisons among $K$ visual generative models, including text-to-image models and text-to-video models. Here, $K > 2$ and is set to 4 in practice. Leveraging the intuitive nature of visual perception, users can confidently vote to rank the outputs based on their preferences. Each data instance $\mathcal{D}^i = (P^i, \mathcal{O}^i, \mathcal{R}^i)$, which consists of one prompt $P^i$ along with the outputs $\mathcal{O}^i = \{O_k^i\}_{k=1}^K$ of the $K$ models $\mathcal{M}^i = \{M_k^i\}_{k=1}^K$ and the user-voted rankings $\mathcal{R}^i = \{R_k^i\}_{k=1}^K$, becomes a preliminary candidate for the dataset $\mathcal{D}_c$. Here, $i = 1, 2, \cdots, N_c$, and $N_c$ is the number of candidate instances.

K-Sort Arena makes efforts in terms of data diversity and voting consistency, thus providing a fundamental assurance of data quality. K-Sort Arena supports input prompts sampled from existing datasets as well as fresh prompts customized by users, which facilitates prompts from diverse domains and varying complexity levels. To ensure the voting quality, all crowdsourced participants are professors and graduate students specializing in visual generation. They all complete pre-voting training, particularly on the evaluation criteria, which is detailed in Appendix A. Additionally, as an open-source project, K-Sort Arena actively encourages contributions from the public community, with the criteria serving as a guiding reference for their voting as well.

To date, K-Sort Arena has collected thousands of votes from both crowdsourced participants and the public community. For text-to-image generation, we have gathered over 1,800 human votes across 35 models, resulting in more than 10,800 pairwise comparisons. For text-to-video generation, we have collected more than 700 human votes across 14 models, which are equivalent to more than 4,200 pairwise comparisons. However, despite training and provided guidelines, inherent subjective differences among individuals can lead to inconsistencies in voting, with some votes deviating from typical preference patterns. In some cases, unintended operational errors may further introduce inaccuracies, posing the risk of preference data contamination.

To this end, we apply a careful filtering for each instance to ensure a representative dataset. Due to probabilistic modeling and Bayesian updating, the leaderboard constructed by K-Sort Arena demonstrates strong robustness to preference noise, i.e., the leaderboard is sufficiently reliable. Thus, we use the consistency between the local ranking $\mathcal{R}^i$ within each instance and the overall ranking $\mathcal{R}^{(L)}$ in the leaderboard as the filtering criterion. Specifically, we quantify this consistency by calculating Spearman's rank correlation coefficient $\rho$ as follows:

$$\rho_i = \frac{\sum_{k=1}^K \left( R_k^i - \bar{R}^i \right)\left( R_k^{(L)} - \bar{R}^{(L)} \right)}{\sqrt{\sum_{k=1}^K \left( R_k^i - \bar{R}^i \right)^2} \cdot \sqrt{\sum_{k=1}^K \left( R_k^{(L)} - \bar{R}^{(L)} \right)^2}} \tag{1}$$

where $R_k^i$ denotes the local ranking assigned to model $M_k^i$, and $R_k^{(L)}$ denotes the corresponding ranking in the global leaderboard $\mathcal{R}^{(L)}$ of the same model $M_k^i$ in $\mathcal{R}^i$. $\bar{R}^i$ and $\bar{R}^{(L)}$ are their respective mean rankings. With the coefficient $\rho$, the filtered dataset is obtained as follows:

$$\mathcal{D} = \{\mathcal{D}^i \mid \rho_i > \tau, \mathcal{D}^i \in \mathcal{D}_c, i = 1, 2, \cdots, N_c\} \tag{2}$$

where $\tau$ is the filtering threshold. Threshold selection is presented in Appendix B. Furthermore, we apply Llama Guard (Inan et al., 2023) to identify and filter out user prompts that are potentially harmful or offensive, which ensures the exclusion of inappropriate content, contributing to the creation of a dataset that is broadly applicable and ethically sound.

Following the curation and filtering processes outlined above, the K-Sort Eval dataset is ultimately established. The dataset description, including size and format, is presented in Table 2.

Table 2: Description of the curated dataset.

| Model | #Instance | #Visual Data | Visual Format | Annotation |
|---|---|---|---|---|
| Text-to-Image | 500 | 2,000 | 512×512 | [1,2,3,4] |
| Text-to-Video | 300 | 1,200 | 512×512, 8 FPS, 5s | [1,2,3,4] |

## 3.2 POSTERIOR CORRECTION FOR EVALUATION RELIABILITY

In order to align with K-Sort Arena (Li et al., 2025b), we follow the probabilistic modeling approach for model capability $\theta$ as follows:

$$\theta \sim \mathcal{N}(\mu, \sigma^2) \tag{3}$$

where $\mu$ and $\sigma$ are the model's expected capability and uncertainty, respectively, and $\mathcal{N}(\cdot)$ denotes the normal distribution. In each round of voting, the probability density of the model's current capability $P(\theta)$ is taken as the prior probability, while the voting result $P(D|\theta)$ serves as the likelihood function for the observation $D$ conditioned on $\theta$. The posterior distribution of the capability $P(\theta|D)$ is then computed using Bayes' theorem as follows:

$$P(\theta|D) = \frac{P(D|\theta)P(\theta)}{\int_{-\infty}^{\infty} P(D|\theta')P(\theta')d\theta'} = \frac{P(D|\theta)P(\theta)}{C} \tag{4}$$

With the posterior probability, the posterior mean and variance of the model capability are updated as follows:

$$\hat{\mu} = \mathbb{E}[\theta|D] = \int_{-\infty}^{\infty} \theta P(\theta|D)d\theta$$
$$\hat{\sigma}^2 = \text{Var}[\theta|D] = \int_{-\infty}^{\infty} (\theta - \mathbb{E}[\theta|D])^2 P(\theta|D)d\theta \tag{5}$$

The derivation and results of the above equations are detailed in Appendix C.

**Posterior Correction.** The above is the updating process under the ideal condition of unbiased human preferences. However, when employing a VLM-as-a-Judge, the results inherently contain hallucinations and biases, which cannot ensure alignment with true human preferences. We define the misalignment between VLM predictions and human preferences as observation noise, and accordingly model the conditional distribution of the observation as a mixture distribution, resulting in the following noise-aware likelihood function:

$$\widetilde{P}(D|\theta) = \lambda P(D|\theta) + (1 - \lambda)P_n(D) \tag{6}$$

where $P_n(D)$ is the noise distribution of observation $D$, and $\lambda \in [0, 1]$ is the confidence coefficient of the observation, with $\lambda = 1$ indicating perfect reliability and no noise.

**Assumption 1.** *Assume that $P_n(D)$, representing a non-informative noise distribution over the observation $D$, is statistically independent of the parameter $\theta$.*

**Lemma 1.** *Under Assumption 1, when the observation is subject to contamination by the noise distribution $P_n(D)$, the resulting posterior distribution $\widetilde{P}(\theta|D)$ can be represented as a mixture of the noise-free posterior distribution and the prior distribution. Specifically, it holds that:*

$$\widetilde{P}(\theta|D) = \lambda' P(\theta|D) + (1 - \lambda')P(\theta) \tag{7}$$

*where $\lambda' \in [0, 1]$ reflects the relative credibility of the posterior distribution induced by the observation with respect to the prior.*

*Proof.* According to Bayes' theorem, when the likelihood function is computed as in Eq. 6, the posterior probability is given by:

$$\widetilde{P}(\theta|D) = \frac{\widetilde{P}(D|\theta)P(\theta)}{\int_{-\infty}^{\infty} \widetilde{P}(D|\theta')P(\theta')d\theta'} = \frac{[\lambda P(D|\theta) + (1-\lambda)P_n(D)]P(\theta)}{\int_{-\infty}^{\infty}[\lambda P(D|\theta') + (1-\lambda)P_n(D)]P(\theta')d\theta'} \tag{8}$$

Based on the additivity of integration, the expression in the denominator can be split into two separate terms, $\int_{-\infty}^{\infty} \lambda P(D|\theta')P(\theta')d\theta'$ and $\int_{-\infty}^{\infty}(1-\lambda)P_n(D)P(\theta')d\theta'$. Base on the homogeneity, the first term can be simplified as:

$$\int_{-\infty}^{\infty} \lambda P(D|\theta')P(\theta')d\theta' = \lambda \int_{-\infty}^{\infty} P(D|\theta')P(\theta')d\theta' = \lambda C \tag{9}$$

where $C$ is the normalizing constant as in Eq. 4. For the second item, with Assumption 1, since $P_n(D)$ is independent of $\theta$, we have:

$$\int_{-\infty}^{\infty} (1-\lambda)P_n(D)P(\theta')\, d\theta' = (1-\lambda)P_n(D)\int_{-\infty}^{\infty} P(\theta')\, d\theta' = (1-\lambda)P_n(D) \qquad (10)$$

Substituting the simplified terms into the expression for the posterior, we obtain:

$$\begin{aligned}
\widetilde{P}(\theta|D) &= \frac{[\lambda P(D|\theta) + (1-\lambda)P_n(D)]P(\theta)}{\lambda C + (1-\lambda)P_n(D)} \\
&= \frac{\lambda C}{\lambda C + (1-\lambda)P_n(D)}\frac{P(D|\theta)P(\theta)}{C} + \frac{(1-\lambda)P_n(D)}{\lambda C + (1-\lambda)P_n(D)}P(\theta)
\end{aligned} \qquad (11)$$

When the actual observation is $D^*$, we define $\lambda' = \lambda C/[\lambda C + (1-\lambda)P_n(D^*)]$. This completes the proof of Lemma 1. □

According to Lemma 1, the posterior under noise can be viewed as a weighted combination between the noise-free posterior and the prior. To derive the weighting factor, we treat human preferences $\mathcal{R}^i$ in the dataset as supervision, and quantify the noise level in the VLM outputs by computing Spearman's rank correlation coefficient $\rho'$ as follows:

$$\rho'_i = \frac{\sum_{k=1}^{K}\left(R_k^{(\text{VLM})} - \bar{R}^{(\text{VLM})}\right)\left(R_k^i - \bar{R}^i\right)}{\sqrt{\sum_{k=1}^{K}\left(R_k^{(\text{VLM})} - \bar{R}^{(\text{VLM})}\right)^2} \cdot \sqrt{\sum_{k=1}^{K}\left(R_k^i - \bar{R}^i\right)^2}} \qquad (12)$$

where $R_k^{(\text{VLM})}$ denotes the VLM's ranking result of the same model $M_k^i$ in $\mathcal{R}^i$. Here, the range of $\rho'$ is [-1,1]. To further normalize it and constrain the values to [0,1] as in Eq. 7, we apply the sigmoid function as follows:

$$\lambda'_i = \text{Sigmoid}(\kappa\rho'_i) = \frac{1}{1+e^{-\kappa\rho'_i}} \qquad (13)$$

where $\kappa$ is the coefficient that controls the slope.

Given $\lambda'$, we proceed to derive the posterior mean and variance under the presence of noise. In Eq. 7, the weighted posterior $\lambda'P(\theta|D)$ follows a normal distribution $\mathcal{N}(\lambda'\hat{\mu}, \lambda'^2\hat{\sigma}^2)$, and the weighted prior $(1-\lambda')P(\theta)$ follows a normal distribution $\mathcal{N}((1-\lambda')\mu, (1-\lambda')^2\sigma^2)$. Since $P(\theta|D)$ and $P(\theta)$ are independently distributed, the additive property of normal distributions applies. Therefore, their weighted sum $\widetilde{P}(\theta|D)$ also follows a normal distribution, with its mean and variance given by:

$$\begin{aligned}
\hat{\mu}_c &= \lambda'\hat{\mu} + (1-\lambda')\mu \\
\hat{\sigma}_c^2 &= \lambda'^2\hat{\sigma}^2 + (1-\lambda')^2\sigma^2
\end{aligned} \qquad (14)$$

where $\hat{\mu}_c$ and $\hat{\sigma}_c^2$ are the corrected posterior mean and variance, respectively.

### 3.3 Dynamic Matching for Evaluation Efficiency

Modern datasets tend to establish their authority through increasingly large scales. However, traditional evaluation methods predominantly rely on static evaluation, which requires exhaustively traversing all instances in the dataset, regardless of the model characteristics or the task complexity. This uniform strategy typically incurs numerous low-gain and unnecessary processes, potentially leading to redundant computation and inefficient evaluation (Kossen et al., 2021; Polo et al., 2024).

Therefore, we are motivated to adaptively select a representative subset based on model-specific traits, enabling a more efficient evaluation process. Thanks to the probabilistic modeling of model capabilities, the evaluation process is equipped with a clear stopping criterion, i.e., the capability uncertainty $\sigma$ reaches the predefined threshold. To this end, we propose a dynamic matching strategy, aiming to dynamically select the dataset instance that is expected to make the largest reduction in the current uncertainty. Specifically, we introduce an uncertainty criterion and a diversity criterion to jointly guide the selection process, thereby maximizing the benefit of each comparison.

**Uncertainty Criterion.** Traditional approaches, such as exhaustive or random matching, can lead to uninformative comparisons. For instance, even when the current model has a high confidence in

achieving a strong capability score, it may still be matched against a significantly weaker model. Such comparisons yield limited gains for updating the model capability. Therefore, our goal is to promote matchups between models of comparable strength. In this way, the model maintains approximately a 50% win rate, indicating maximum uncertainty in the comparison outcome. Based on this insight, we define the uncertainty criterion $U_{\text{unc}}$ as follows:

$$U_{\text{unc}}^i = -\frac{1}{K} \sum_{k=1}^{K} \left| \mu - \mu_k^i \right| \tag{15}$$

where $\mu$ is the current capability mean of the new model being evaluated, and $\mu_k^i$ is the capability mean of the $k$-th model in the $i$-th dataset instance.

**Diversity Criterion.** In the proposed dataset, there are $K$ models in each instance, making the evaluation of a new model essentially a one-to-many matching process. This requires not only considering the relationship between the new model and each model within the instance, but also accounting for the interrelations among the $K$ models themselves. To this end, we aim to ensure that the group covers as diverse a set of opponents as possible, thereby avoiding homogeneous matchups and reducing information redundancy. Here, we quantify the diversity among models within an instance by measuring the degree of overlap between their Gaussian-modeled capability distributions, and the diversity criterion $U_{\text{div}}$ is defined as follows:

$$U_{\text{div}}^i = -\sum_{1 \le k_1 < k_2 \le K} \int_{-\infty}^{\infty} \min\left(P(\theta_{k_1}^i), P(\theta_{k_2}^i)\right) d\theta \tag{16}$$

where $P(\theta_k^i)$ is the probability density of the $k$-th model's capability in the $i$-th dataset instance.

Based on the above two criterions, we can dynamically match the next dataset instance by maximizing the expected gain with respect to the current model status as follows:

$$i^* = \arg\max_i \left(U_{\text{unc}}^i + \alpha U_{\text{div}}^i\right) \tag{17}$$

where $\alpha$ is a balancing coefficient.

## 3.4 OVERALL PIPELINE OF K-SORT EVAL

In this section, we present the overall pipeline of K-Sort Eval in evaluating a new model. Specifically, we first initialize its capability $(\mu, \sigma)$ and then put it into the following procedures:

▷ **Dynamic Matching**: We select a dataset instance using Eq. 17, and form a group of size $K{+}1$ by combining the new model with those in the selected instance.

▷ **VLM Judgement**: To mitigate hallucinations of the VLM, we adopt two prompt design strategies: swapping operation and rule augmentation. Specifically, we first randomly shuffle the $K{+}1$ models to eliminate potential positional biases. Then, following the voting criteria in K-Sort Arena (Li et al., 2025b), we provide the VLM with identical judgement instructions, as presented in Appendix D.

▷ **Updating with Correction**: We compute the posterior mean and variance under a noise-free assumption via Eq. 5, and subsequently correct the results using Eq. 14.

The above procedures are iteratively executed until the value of $\sigma$ falls below a predefined stopping criterion. Finally, the model capability is estimated using the conservative score (Phillips & Edwards, 1966) defined as $S = \mu - \eta\sigma$, where $\eta$ is a coefficient typically set to 3.0.

## 4 EXPERIMENTS

### 4.1 EXPERIMENTAL SETUP

K-Sort Eval enables automated preference evaluation of new models. For the VLM selection, GPT-4o (OpenAI, 2024) is used as the judge for images, while Qwen-VL-Max (Bai et al., 2023a) is used for videos[1]. It quantifies model capability using both absolute and relative metrics: the absolute metric is the conservative score, while the relative metric is the model's ranking in K-Sort Arena. The

---

[1]GPT-4o API does not natively support video input.

Table 3: Validation of evaluation reliability of K-Sort Eval for text-to-image/video models. The model scores and rankings produced by K-Sort Eval are highly consistent with K-Sort Arena.

| *Text-to-Image* | FLUX.1-dev | | Midjourney-v5.0 | | Realvisxl-v3.0 | | Dalle-2 | | SD-v1.5 | |
|---|---|---|---|---|---|---|---|---|---|---|
| | Rank | Score | Rank | Score | Rank | Score | Rank | Score | Rank | Score |
| K-Sort Arena | 5 | 28.83 | 11 | 27.44 | 16 | 23.93 | 24 | 21.74 | 29 | 20.10 |
| K-Sort Eval (Ours) | 5 | 28.86 | 11 | 27.50 | 16 | 24.02 | 24 | 21.79 | 29 | 20.03 |

| *Text-to-Video* | Runway-Gen3 | | CogVideoX-5b | | KLing-v1.0 | | Pika-v1.0 | | VideoCrafter2 | |
|---|---|---|---|---|---|---|---|---|---|---|
| | Rank | Score | Rank | Score | Rank | Score | Rank | Score | Rank | Score |
| K-Sort Arena | 2 | 33.93 | 3 | 33.60 | 5 | 32.80 | 7 | 29.17 | 12 | 23.65 |
| K-Sort Eval (Ours) | 2 | 33.98 | 3 | 33.63 | 5 | 32.90 | 7 | 29.10 | 12 | 23.72 |

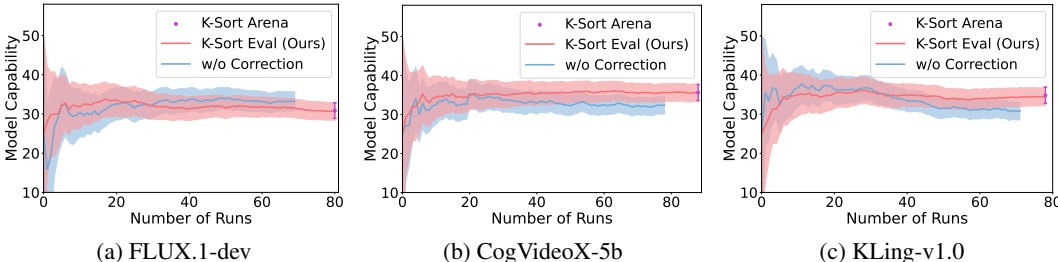

(a) FLUX.1-dev        (b) CogVideoX-5b        (c) KLing-v1.0

Figure 2: Visualization of the evaluation processes. With posterior correction, K-Sort Eval achieves a smoother trajectory and produces more accurate results that are consistent with K-Sort Arena.

referenced Arena leaderboards are the version updated on Sep 15, 2025, as presented in Appendix E. In addition to reliability, K-Sort Eval also offers a notable efficiency advantage, as measured by the number of model runs required for evaluation, which equals the number of VLM calls. For dataset curation, we set the filtering threshold $\tau$ to 0.75. The $\sigma$ threshold in the stopping criterion is set to 0.75. The coefficients $\kappa$ and $\alpha$ are set to 5.0 and 0.5, respectively, after a simple grid search.

## 4.2 VALIDATION OF EVALUATION RELIABILITY AND EFFICIENCY

**Evaluation Reliability of K-Sort Eval.** We select models from the K-Sort Arena (Li et al., 2025b) leaderboard and evaluate them using K-Sort Eval, including text-to-image and text-to-video models. These models span different positions on the leaderboard to demonstrate the generalizability of our dataset and method. The results, including both rankings and scores, are compared with those in K-Sort Arena, as shown in Table 3. The evaluation results of K-Sort Eval are consistent with those of K-Sort Arena, which is entirely based on human preferences. For instance, in the evaluation of FLUX.1-dev, the score produced by K-Sort Eval differs by only 0.03 from that in K-Sort Arena, with both methods assigning it the same rank of 5, which highlights the effectiveness of K-Sort Eval.

**Evaluation Efficiency of K-Sort Eval.** Thanks to the proposed dynamic matching strategy, the evaluation process does not requires traversing the entire dataset, which significantly improves efficiency. Figure 3 illustrates the number of runs required for the new model, with data from 100 tries covering all models. The vast majority (91% for images, 93% for videos) complete the evaluation in less than 90 runs, which is a significant efficiency gain over existing methods such as FID (typically 50,000 runs) (Heusel et al., 2017), GenAI-Bench (1,600 runs) (Li et al., 2024a).

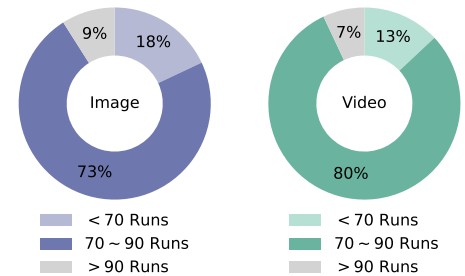

Figure 3: Number of runs required for the new model in the evaluation.

### 4.3 COMPARISON WITH PREFERENCE SCORING METHODS

We select 100 instance groups to compare the correlations between different methods and actual human preferences, including CLIP-based scoring methods (ImageReward (Xu et al., 2023), PickScore (Kirstain et al., 2023), HPS (Wu et al., 2023), and VQAScore (Li et al., 2024a)) and VLM-based methods using GPT-4o (OpenAI, 2024), as shown in Figure 4. We also report the result with correction obtained through $\lambda'$ weighting. GPT-4o consistently outperforms CLIP-based methods, and introducing correction significantly improves overall correlation, as it reduces the influence of noisy observations.

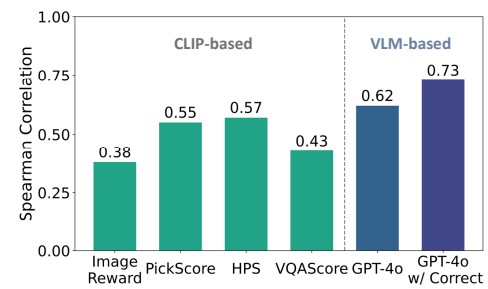

Figure 4: Correlations of different methods with actual human preferences.

### 4.4 APPLICATION OF EVALUATION ON COMPRESSED MODELS

The evaluation of compressed models is also crucial in real-world applications (Li & Gu, 2023; Li et al., 2023c; 2022). K-Sort Eval provides both absolute scores and relative rankings, thus it not only assesses the performance degradation after compression, but also identifies which standard model the compressed model is functionally comparable to.

Table 4: Application of evaluation on compressed models, which reveals not only changes in absolute scores, but also the results of relative rankings. The model size is calculated in FP16 by default.

(a) Distilled models

| Model | Rank | Score | Size (GB) | Step |
|---|---|---|---|---|
| SD-v3.5-large | 3 | 28.95 | 16.2 | 40 |
| Dalle-3 | 8 | 28.25 | - | - |
| SD-v3.5-large-turbo | 9 | 27.71 | 16.2 | 4 |
| FLUX.1-schnell | 10 | 27.69 | 24 | 4 |
| SDXL | 32 | 18.85 | 5.2 | 25 |
| SD-v1.5 | 29 | 20.11 | 1.72 | 50 |
| SDXL-SSD-1b | 30 | 19.40 | 2.6 | 25 |
| SD-v2.1 | 31 | 18.95 | 1.73 | 50 |

(b) Quantized models

| Model | Rank | Score | Size (GB) | Step |
|---|---|---|---|---|
| FLUX.1-dev | 5 | 28.83 | 24 | 28 |
| Dalle-3 | 8 | 28.27 | - | - |
| NF4 (BNB) | 9 | 27.93 | 6 | 28 |
| SD-v3.5-large-turbo | 9 | 27.73 | 16.2 | 4 |
| FLUX.1-schnell | 10 | 27.71 | 24 | 4 |
| W4A4 (SVDQuant) | 11 | 27.66 | 6 | 28 |
| Midjourney-v5.0 | 11 | 27.44 | - | - |

**Distilled Models.** Table 4a gives examples of distilled models, with reduced step and model size, respectively. For SD-v3.5-large-turbo, the number of inference steps is reduced from 40 to 4, resulting in a score drop of 1.24 and a ranking shift from 4 to 9. Based on its ranking, we easily conclude that its performance is comparable to that of Dalle-3 and FLUX.1-schnell.

**Quantized Models.** Table 4b reports the quantization results of FLUX.1-dev, including BNB (Dettmers et al., 2023) and SVDQuant (Li et al., 2024c). When quantizing in NF4 format, the model size is reduced by 4×, while delivering a score decrease of 0.90. Crucially, it offers an intuitive measure of relative capability and directly points to a benchmark model of similar strength.

### 4.5 ABLATION STUDIES

**Effect of the Proposed Modules.** We verify the validity of posterior correction and dynamic matching, as shown in Table 5. When evaluating FLUX.1-dev, without posterior correction, every VLM judgment is fully accepted, even when misaligned with human preferences. This leads to reduced evaluation accuracy, with a score deviation of 0.49 and a rank discrepancy of 2 compared to Arena. Additionally, in the absence of dynamic matching, the entire dataset needs to be traversed, leading to increased costs. The grid search of coefficients $\kappa$ and $\alpha$ are shown in Appendix F.

**Prompt Designs for VLM.** Table 5 further illustrates the impact of prompt designs. In the case of FLUX.1-dev, for example, the model ranking is shifted when the wrapping operation is removed.

Table 5: Ablation studies on effect of the proposed modules and prompt designs. We report the results of text-to-image model FLUX.1-dev and text-to-video model CogVideoX-5b.

<table>
<tr><td colspan="4" align="center">(a) FLUX.1-dev</td><td colspan="4" align="center">(b) CogVideoX-5b</td></tr>
<tr><td>Method</td><td>Rank</td><td>Score</td><td>#Runs</td><td>Method</td><td>Rank</td><td>Score</td><td>#Runs</td></tr>
<tr><td>K-Sort Arena</td><td>5</td><td>28.83</td><td>-</td><td>K-Sort Arena</td><td>3</td><td>33.60</td><td>-</td></tr>
<tr><td>K-Sort Eval (Ours)</td><td>5</td><td>28.86</td><td>81</td><td>K-Sort Eval (Ours)</td><td>3</td><td>33.63</td><td>89</td></tr>
<tr><td>w/o Posterior Correction</td><td>3</td><td>29.32</td><td>70</td><td>w/o Posterior Correction</td><td>6</td><td>31.86</td><td>79</td></tr>
<tr><td>w/o Dynamic Matching</td><td>5</td><td>28.79</td><td>500</td><td>w/o Dynamic Matching</td><td>3</td><td>33.65</td><td>300</td></tr>
<tr><td>w/o Wapping Operation</td><td>4</td><td>28.93</td><td>79</td><td>w/o Wapping Operation</td><td>3</td><td>33.55</td><td>90</td></tr>
<tr><td>w/o Rule Augmentation</td><td>9</td><td>28.13</td><td>119</td><td>w/o Rule Augmentation</td><td>5</td><td>33.10</td><td>130</td></tr>
</table>

Moreover, without rule augmentation, the VLM lacks clear and uniform principles in the judgment, resulting in a substantial score difference of 0.70.

## 5 CONCLUSION

In this work, we propose K-Sort Eval, a scalable and reliable evaluation framework that leverages vision-language models (VLMs) with posterior correction and dynamic matching strategies to approximate human preferences in generative model assessment. By utilizing high-quality dataset from K-Sort Arena and introducing Bayesian correction based on VLM-human consistency, K-Sort Eval significantly improves alignment with human judgements. Furthermore, the proposed dynamic matching enhances evaluation efficiency by selecting instances with maximum expected gains. Experimental results show that K-Sort Eval achieves alignment with human-voted scores and rankings, while substantially reducing evaluation costs, highlighting its reliability and efficiency.

## ACKNOWLEDGMENTS

This work is supported in part by the Strategic Priority Research Program of Chinese Academy of Sciences under Grant Number XDB1100000; in part by the National Natural Science Foundation of China under Grant Number 62276255; in part by the Postdoctoral Fellowship Program of CPSF under Grant Number GZC20251175. Yang You's research group is being sponsored by NUS startup grant (Presidential Young Professorship), Singapore MOE Tier-1 grant, ByteDance grant, NUS ARTIC grant, Apple grant, Alibaba grant and Adobe gift.

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

## A EVALUATION CRITERIA IN K-SORT ARENA

In K-Sort Arena, all crowdsourced participants are professors and graduate students specializing in visual generation, affiliated with institutions such as University of California Berkeley, Chinese Academy of Sciences, National University of Singapore, and Nanyang Technological University, etc. They all complete pre-voting training, particularly on the following evaluation criteria:

▷ **Text-to-Image Models**  The evaluation is based on alignment (50%) and aesthetics (50%). Alignment encompasses entity (30%) and style (20%), while aesthetics includes photorealism (30%), light and shadow rendering (10%), and the absence of artifacts (10%).

▷ **Text-to-Video Models**  The models are also evaluated based on alignment (50%) and aesthetics (50%). Alignment is assessed based on video content matching (20%), movement matching (15%), and inter-frame consistency (15%), while aesthetics considers photorealism (30%), physical correctness (10%), and the absence of artifacts (10%).

Additionally, as an open-source project, K-Sort Arena actively encourages contributions from the public community, with the criteria serving as a guiding reference for their voting as well.

## B FILTERING THRESHOLD IN DATASET CURATION

We filter the data by the Spearman's rank correlation coefficient between the local rankings within the dataset and the corresponding model's ranking in the overall leaderboard to prevent preference contamination. Due to performance fluctuations relative to a model's true capability and the presence of ties, even preference-aligned data cannot always guarantee a correlation of 1.0. Therefore, it is necessary to determine a sufficiently reliable selection threshold.

Table 6 lists spearman's rank correlation coefficients in different cases, including tie and misordering cases. In our dataset curation, we consider the cases of tie between two models and misordering between two models to be valid samples, while the cases of misordering among three models is invalid samples. As a result, in order to balance validity and diversity, we set the filtering threshold to 0.75.

Table 6: Spearman's rank correlation coefficients in different cases, including tie and misordering.

| Case | Rank | Spearman |
|------|------|----------|
| Ground Truth | [0,1,2,3] | - |
| Fully consistent | [0,1,2,3] | 1.00 |
| Tie between two models | [0,1,1,2] | 0.95 |
| Tie among three models | [0,1,1,1] | 0.77 |
| Misordering between two models | [0,2,1,3] | 0.80 |
| Misordering among three models | [0,3,1,2] | 0.40 |

## C DERIVATION OF BAYESIAN UPDATING

We begin by analyzing the case of two competing models, $M_1$ and $M_2$, before generalizing to the comparison among $K$ models. Suppose the observation $D$ indicates that model $M_1$ outperforms model $M_2$. The likelihood of this event, conditioned on the latent performance parameters $\theta_1$ and $\theta_2$, is given by:

$$P(D|\theta_1,\theta_2) = P(X_1 > X_2) = \Phi\left(\frac{\theta_1 - \theta_2}{\sqrt{\beta_1^2 + \beta_2^2}}\right) \qquad (18)$$

where $\Phi(x)$ denotes the cumulative distribution function (CDF) of the standard normal distribution, and $\phi(x)$ is the corresponding probability density function (PDF):

$$\Phi(x) = \int_{-\infty}^{x} \phi(u)\,du, \quad \phi(x) = \frac{1}{\sqrt{2\pi}} e^{-x^2/2} \qquad (19)$$

Using Bayes' theorem, we can then derive the joint posterior distribution of $(\theta_1, \theta_2)$ given the observation $D$ as follows:

$$P(\theta_1,\theta_2|D) \propto P(\theta_1)P(\theta_2)P(D|\theta_1,\theta_2) = \phi\left(\frac{\theta_1 - \mu_1}{\sigma_1}\right)\phi\left(\frac{\theta_2 - \mu_2}{\sigma_2}\right)\Phi\left(\frac{\theta_1 - \theta_2}{\sqrt{\beta_1^2 + \beta_2^2}}\right) \quad (20)$$

The marginal posterior distribution of $\theta_1$ can be obtained by integrating out $\theta_2$ from the joint posterior:

$$P(\theta_1|D) = \int_{-\infty}^{\infty} P(\theta_1, \theta_2|D)\, d\theta_2 \propto \phi\left(\frac{\theta_1 - \mu_1}{\sigma_1}\right) \Phi\left(\frac{\theta_1 - \mu_2}{\sqrt{\beta_1^2 + \beta_2^2 + \sigma_2^2}}\right) \tag{21}$$

Given the marginal posterior distribution, the posterior expectation of $\theta_1$ can then be computed as:

$$\hat{\mu}_1 = E\left[\theta_1|D\right] = \frac{\int_{-\infty}^{\infty} \theta_1 P(\theta_1|D)d\theta_1}{\int_{-\infty}^{\infty} P(\theta_1|D)d\theta_1} = \mu_1 + \frac{\sigma_1^2}{\sqrt{\sum\left(\beta_i^2 + \sigma_i^2\right)}} \frac{\phi\left(\frac{\mu_1 - \mu_2}{\sqrt{\sum\left(\beta_i^2 + \sigma_i^2\right)}}\right)}{\Phi\left(\frac{\mu_1 - \mu_2}{\sqrt{\sum\left(\beta_i^2 + \sigma_i^2\right)}}\right)} \tag{22}$$

$$= \mu_1 + \frac{\sigma_1^2}{c_{12}} \cdot \mathcal{V}\left(\frac{\mu_1 - \mu_2}{c_{12}}\right)$$

where $\mathcal{V}(x) = \phi(x)/\Phi(x)$ and $c_{ij}^2 = \sum\left(\beta_i^2 + \sigma_i^2\right)$. The mean $\hat{\mu}_1$ of $\theta_1$ is updated accordingly based on the observed outcome. In a similar manner, the posterior variance $\hat{\sigma_1}^2$ is updated using the following expression:

$$\hat{\sigma_1}^2 = Var[\theta_1|D] = E[\theta_1^2|D] - (E[\theta_1|D])^2 = \sigma_1^2 \cdot \left(1 - \frac{\sigma_1^2}{\sum\left(\beta_i^2 + \sigma_i^2\right)} \cdot \mathcal{W}\left(\frac{\mu_1 - \mu_2}{\sqrt{\sum\left(\beta_i^2 + \sigma_i^2\right)}}\right)\right)$$

$$= \sigma_1^2 \cdot \left(1 - \frac{\sigma_1^2}{c_{12}^2} \cdot \mathcal{W}\left(\frac{\mu_1 - \mu_2}{c_{12}}\right)\right) \tag{23}$$

where $\mathcal{W}(x) = \mathcal{V}(x)(\mathcal{V}(x) + x)$. The aforementioned process completes the Bayesian updating for a pairwise comparison between two models. We now extend this framework to a free-for-all comparison among $K$ models. In this case, the update rules for the performance parameters of the $i$-th model are given by the following equations:

$$\hat{\mu}_i = \mu_i + \sigma_i^2 \cdot \left(\sum_{q:r_q > r_i} \frac{1}{c_{iq}} \cdot \mathcal{V}\left(\frac{\mu_i - \mu_q}{c_{iq}}\right) + \sum_{q:r_i < r_q} \frac{-1}{c_{iq}} \cdot \mathcal{V}\left(\frac{\mu_q - \mu_i}{c_{iq}}\right)\right) \tag{24}$$

$$\hat{\sigma_i}^2 = \sigma_i^2 \cdot \left(1 - \left(\sum_{q:r_i > r_q} \frac{\sigma_i^2}{c_{iq}^2} \cdot \mathcal{W}\left(\frac{\mu_i - \mu_q}{c_{iq}}\right) + \sum_{q:r_i < r_q} \frac{\sigma_i^2}{c_{iq}^2} \cdot \mathcal{W}\left(\frac{\mu_q - \mu_i}{c_{iq}}\right)\right)\right) \tag{25}$$

## D   RULE AUGMENTATION FOR VLM PROMPT

We adopt the rule augmentation strategy to provide clear and effective guidance for VLM judgements. To ensure consistency with K-Sort Arena, these rules are aligned with the manual voting criteria used by human annotators. This enhances the interpretability of VLM outputs and improves their comparability with human preferences. The complete prompt design is illustrated in Figure 5.

---

**VLM Prompt for Text-to-Image Models**

You will be given 5 images. Your task is to rank them from best to worst based on **visual aesthetics** (50%) and **alignment** with the given description (50%): **[prompt]**.
- **Aesthetics** includes photorealism (30%), light and shadow (10%), and absence of artifacts (10%);
- **Alignment** includes entity matching (30%) and style matching (20%);
You MUST respond with only the sorted images in the strict format: "Image 1, Image 2, Image 3, Image 4, Image 5".

**VLM Prompt for Text-to-Video Models**

You will be given 5 videos. Your task is to rank them from best to worst based on **visual aesthetics** (50%) and **alignment** with the given description (50%): **[prompt]**.
- **Aesthetics** includes photorealism (30%), physical correctness (10%), and absence of artifacts (10%);
- **Alignment** includes content matching (20%), movement matching (15%), and inter-frame consistency (15%);
You MUST respond with only the sorted videos in the strict format: "video 1, video 2, video 3, video 4, video 5".

---

Figure 5: Prompt design that provides voting criteria consistent with human voting in K-Sort Arena, serving as guidance for the VLM Judgement and helping reduce hallucinations.

# E    K-SORT ARENA LEADERBOARD

We use the leaderboard provided by K-Sort Arena (Li et al., 2025b) as the ground-truth baseline for human preferences. In this work, all referenced leaderboards are based on the version updated on Sep 15, 2025, as shown in Table 7.

Table 7: K-Sort Arena leaderboards updated on Sep 15, 2025.

(a) Text-to-Image Models

| Rank | Model | Organization | Score ($\mu/\sigma$) |
|---|---|---|---|
| 1 | GPT-4o | OpenAI | 30.86 (33.20 / 0.78) |
| 2 | FLUX-1.1-pro | Black Forest Labs | 29.52 (31.57 / 0.68) |
| 3 | SD-v3.5-large | Stability AI | 28.97 (31.13 / 0.72) |
| 4 | FLUX.1-pro | Black Forest Labs | 28.90 (30.89 / 0.66) |
| 5 | FLUX.1-dev | Black Forest Labs | 28.83 (30.81 / 0.66) |
| 6 | Aurora | xAI | 28.72 (31.05 / 0.78) |
| 7 | Midjourney-v6.0 | Midjourney | 28.64 (30.64 / 0.67) |
| 8 | Dalle-3 | OpenAI | 28.27 (30.26 / 0.67) |
| 9 | SD-v3.5-large-turbo | Stability AI | 27.73 (29.94 / 0.74) |
| 10 | FLUX.1-schnell | Black Forest Labs | 27.71 (29.72 / 0.67) |
| 11 | Midjourney-v5.0 | Midjourney | 27.44 (29.47 / 0.68) |
| 12 | SD-v3.0 | Stability AI | 27.13 (29.10 / 0.66) |
| 13 | Pixart-Sigma | PixArt-Alpha | 26.38 (28.39 / 0.67) |
| 14 | Proteus-v0.2 | DataAutoGPT3 | 24.69 (26.68 / 0.67) |
| 15 | Open-Dalle-v1.1 | DataAutoGPT3 | 24.65 (26.65 / 0.67) |
| 16 | Realvisxl-v3.0 | Realistic Vision | 23.93 (25.94 / 0.67) |
| 17 | Dreamshaper-xl | Lykon | 23.89 (25.85 / 0.66) |
| 18 | Realvisxl-v2.0 | Realistic Vision | 23.87 (25.87 / 0.67) |
| 19 | Kandinsky-v2.2 | AI-Forever | 23.57 (25.56 / 0.66) |
| 20 | Deepfloyd-IF | DeepFloyd | 23.47 (25.47 / 0.67) |
| 21 | Meissonic | Alibaba, Skywork AI | 22.69 (24.93 / 0.75) |
| 22 | Kandinsky-v2.0 | AI-Forever | 22.51 (24.48 / 0.65) |
| 23 | SDXL-turbo | Stability AI | 21.83 (23.93 / 0.70) |
| 24 | Dalle-2 | OpenAI | 21.74 (23.72 / 0.66) |
| 25 | Playground-v2.5 | Playground AI | 21.60 (23.55 / 0.65) |
| 26 | Openjourney-v4 | Prompthero | 21.41 (23.39 / 0.66) |
| 27 | LCM-v1.5 | Tsinghua | 20.89 (22.90 / 0.67) |
| 28 | SD-turbo | Stability AI | 20.25 (22.36 / 0.70) |
| 29 | SD-v1.5 | Stability AI | 20.10 (22.12 / 0.67) |
| 30 | SSD-1b | Segmind | 19.40 (21.41 / 0.67) |
| 31 | SD-v2.1 | Stability AI | 18.94 (20.93 / 0.66) |
| 32 | SDXL | Stability AI | 18.85 (20.84 / 0.66) |
| 33 | Playground-v2.0 | Playground AI | 18.66 (20.67 / 0.67) |
| 34 | SDXL-Lightning | ByteDance | 18.06 (20.05 / 0.67) |
| 35 | Stable-cascade | Stability AI | 16.69 (18.80 / 0.70) |
| 36 | SDXL-Deepcache | NUS | 16.16 (18.15 / 0.66) |

(b) Text-to-Video Models

| Rank | Model | Organization | Score ($\mu/\sigma$) |
|---|---|---|---|
| 1 | Sora (official) | OpenAI | 34.66 (37.42 / 0.92) |
| 2 | Runway-Gen3 | Runway | 33.93 (35.94 / 0.67) |
| 3 | CogVideoX-5b | Tsinghua | 33.60 (35.63 / 0.68) |
| 4 | Sora (release) | OpenAI | 33.53 (35.61 / 0.69) |
| 5 | KLing-v1.0 | Kuaishou | 32.80 (34.84 / 0.68) |
| 6 | Runway-Gen2 | Runway | 29.57 (31.63 / 0.69) |
| 7 | Pika-v1.0 | Pika | 29.17 (31.27 / 0.70) |
| 8 | LaVie | Shanghai AI Lab | 28.68 (30.67 / 0.67) |
| 9 | OpenSora | HPC-AI | 27.39 (29.41 / 0.67) |
| 10 | Pika-beta | Pika | 27.38 (29.49 / 0.70) |
| 11 | AnimateDiff | CUHK etc. | 26.46 (28.49 / 0.68) |
| 12 | VideoCrafter2 | Tencent | 23.65 (25.70 / 0.69) |
| 13 | StableVideoDiffusion | Stability AI | 23.01 (25.09 / 0.70) |
| 14 | Zeroscope-v2-xl | Cerspense | 16.96 (19.33 / 0.79) |

# F    GRID SEARCH OF COEFFICIENTS

We perform a simple grid search of coefficients $\kappa$ and $\alpha$, as reported in Table 8. Based on the grid search results, we select $\kappa = 5$ and $\alpha = 0.5$ as the optimal hyperparameters. This setting achieves a strong alignment with the K-Sort Arena benchmark (rank = 6), while maintaining a competitive performance score (28.86). Notably, it also results in the lowest number of model runs (81), indicating high evaluation efficiency. Compared to other settings, this combination provides the best balance between ranking consistency and evaluation cost.

Table 8: Grid search of coefficients $\kappa$ and $\alpha$. We report results for text-to-image model FLUX.1-dev and they hold for other models.

| $\kappa$ | K-Sort Arena | 1 | 3 | 5 | 7 | 9 |
|---|---|---|---|---|---|---|
| Rank | 6 | 8 | 6 | 6 | 5 | 9 |
| Score | 28.83 | 28.58 | 28.80 | 28.86 | 28.90 | 28.22 |
| #Runs | - | 110 | 90 | 81 | 74 | 72 |

| $\alpha$ | K-Sort Arena | 0.1 | 0.3 | 0.5 | 0.7 | 0.9 |
|---|---|---|---|---|---|---|
| Rank | 6 | 6 | 6 | 6 | 6 | 5 |
| Score | 28.82 | 28.77 | 28.80 | 28.86 | 28.72 | 28.93 |
| #Runs | - | 107 | 89 | 81 | 84 | 90 |

