# OpenReview forum: "K-Sort Eval: Efficient Preference Evaluation for Visual Generation via Corrected VLM-as-a-Judge"
_ICLR.cc/2026/Conference — ICLR 2026 Poster_

### Official Review · Reviewer_9sHD · 2025-10-30

**Soundness:** 2
**Presentation:** 2
**Contribution:** 2
**Rating:** 6
**Confidence:** 3

**Summary:**

This paper proposes K-Sort Eval, a framework for evaluating visual generative models that uses a Vision-Language Model (VLM) as a judge. The goal is to create a more scalable and efficient alternative to human-based "Arena" evaluations. The method leverages a curated, high-quality dataset of human preference rankings from the existing K-Sort Arena. It introduces two key technical contributions: 1) a "posterior correction" method to align VLM judgments with human supervision by modeling VLM-human discrepancies as observation noise, and 2) a "dynamic matching" strategy to efficiently select the most informative comparisons. Experiments show that K-Sort Eval's rankings are consistent with the human-based K-Sort Arena leaderboard, while requiring significantly fewer (under 90) model runs.

**Strengths:**

1. Addresses a Critical Problem: The paper tackles the highly relevant and important challenge of scalable, efficient, and human-aligned evaluation for generative models, which is a major bottleneck in the field.

2. Novel Correction Mechanism: The "posterior correction" method (Section 3.2) is a novel and clever approach to mitigate the known biases and unreliability of VLM-as-a-Judge. Modeling the discrepancy between VLM and human votes as observation noise in a Bayesian framework is a solid technical contribution.

3. Demonstrated Efficiency: The dynamic matching strategy (Section 3.3) and VLM-based approach demonstrably reduce the evaluation cost to fewer than 90 model runs. This is a significant practical improvement over the thousands of comparisons often required in large-scale human evaluations.

**Weaknesses:**

1. Dependency on Curated Dataset: The entire framework seems to be a "closed-world" system. The posterior correction (Eq. 12) appears to require a pre-existing human-voted ranking ($R^i$) for the exact same prompt to calculate the correction factor $\lambda'$. This suggests the method can only be used to evaluate new models on the static, pre-existing prompts (500 for images, 300 for videos) from the curated dataset, which severely limits its generalizability for open-ended evaluation on new, unseen prompts.

2. Potential for "Overfitting" to the Leaderboard: The dataset curation process itself (Section 3.1) filters out human votes that are inconsistent with the overall K-Sort Arena leaderboard. This creates a risk of circularity: the "ground truth" used to build the evaluator is already a product of the system it's meant to replicate. This could cause the evaluator to simply "overfit" to the specific biases of the K-Sort Arena, rather than capturing a more general or robust notion of human preference.

3. Unclear Scalability of Supervision: The reliance on dataset supervision for the correction mechanism is a key weakness. If the method does require human ranking data for every new prompt to be evaluated, it doesn't solve the scalability problem; it just shifts it. If it doesn't require this, the paper fails to explain how the correction $\lambda'$ is calculated for a truly new prompt that isn't in the curated dataset.

4. Limited Ablation on VLM Choice: The choice of VLM (GPT-4o, Qwen-VL-Max) is critical, yet its impact is not deeply explored. The ablation study (Table 5) shows that removing posterior correction dramatically changes the model rankings (e.g., from 3 to 6 for CogVideoX-5b), proving the VLM's raw judgment is unreliable. This makes the system highly dependent on the (unexplored) interaction between the specific VLM and the specific correction dataset.

**Questions:**

1. Could you please clarify the workflow for evaluating a new model on a novel prompt that is not in the curated dataset? How is the posterior correction (Eq. 12) applied in that scenario, given that there is no "dataset supervision" $R^i$ to compare against? Or, is the framework strictly limited to the 800 prompts in the dataset?

2. The dataset curation (Section 3.1) uses a correlation threshold ($\tau=0.75$) to filter out human votes that deviate from the global leaderboard. How did you ensure that this process doesn't just "bake in" the specific biases of the K-Sort Arena leaderboard and remove genuinely diverse or high-variance (but still valid) human preferences?

3. The ablation study (Table 5) shows very large rank changes when posterior correction is removed. This suggests the VLM's "raw" judgment is significantly different from the human-aligned one. Could you provide more analysis on the nature of this discrepancy? For example, is the VLM consistently biased on certain criteria (e.g., aesthetics vs. alignment) or for certain models?

4. How sensitive is the dynamic matching strategy (Eq. 17) to the hyperparameters, particularly the balancing coefficient $\alpha$? The paper states it was set after a simple grid search, but it seems crucial for balancing the exploration of diverse model matchups versus challenging the new model against its closest competitors.

---

> ### Author Response · Authors · 2025-11-19
>
> Thank you for your recognition of our work and your thorough review. Below, We answer all your questions with detailed explanations and clarifications.
>
> ---
>
> > W1&W3&Q1: Dependency on Curated Dataset and Unclear Scalability of Supervision. Could you please clarify the workflow for evaluating a new model on a novel prompt that is not in the curated dataset? How is the posterior correction (Eq. 12) applied in that scenario, given that there is no "dataset supervision" to compare against? Or, is the framework strictly limited to the 800 prompts in the dataset?
> >
>
> Thanks for your insightful comment and question.
>
> We would like to clarify that the K-Sort Eval dataset is **fixed and static**, including the prompts, images, and corresponding rankings. This design is motivated by the following considerations:
> - **Reproducibility and fairness of comparison**:
> As is the case for nearly all existing evaluation datasets, ensuring reproducibility requires the dataset to remain static and the evaluation procedure to be consistent. Without fixing the prompts, it would be hard to guarantee fair comparisons across different users applying the K-Sort Eval metric.
> - **Evaluation efficiency**:
> Each instance in the dataset contains a prompt along with the outputs of K reference models. Thus, when evaluating a new model, one only needs to run the new model; the outputs of the K reference models are already precomputed. If arbitrary new prompts were allowed, all K reference models would have to be re-executed as well, which significantly increases the computational cost.
>
> Moreover, although the K-Sort Eval dataset is fixed and static, we fully agree with you on the **importance of open-ended evaluation on new, unseen prompts**. To this end, we emphasize that K-Sort Eval will **continue to evolve** to strengthen its scalability and generalizability. Specifically, the K-Sort Eval dataset is derived from K-Sort Arena, where users can freely submit new prompts. As K-Sort Arena continuously collects user-generated data, we promise to **periodically release updated versions** of K-Sort Eval (V1, V2, V3, ...), incorporating an expanding pool of prompt candidates, which ensures that K-Sort Eval remains up-to-date.
>
>
> In addition, although our current setup restricts K-Sort Eval to the static prompts in its dataset, we further analyzed, following your suggestion, the feasibility of **performing posterior correction with new prompts**. For a new prompt, both the new model and the K matched reference models would need to run inference. Since the outputs of these K models would no longer have dataset-level supervision, a potential approach is to directly use the leaderboard ranks as the supervision signal. We would like to clarify, however, that this is only a preliminary possibility; because such supervision may introduce accuracy concerns, we plan to investigate it more thoroughly in future work. In the current version of K-Sort Eval, considering efficiency, supervision reliability, and reproducibility, we adopt a static prompt dataset.
>
>
> > W2: Potential for "Overfitting" to the Leaderboard: The dataset curation process itself (Section 3.1) filters out human votes that are inconsistent with the overall K-Sort Arena leaderboard. This creates a risk of circularity. This could cause the evaluator to simply "overfit" to the specific biases of the K-Sort Arena, rather than capturing a more general or robust notion of human preference.
> >
>
> Thanks for your great comment. As you noted, the goal of K-Sort Eval is to approximate the K-Sort Arena. Therefore, the validity of K-Sort Eval fundamentally relies on the reliability of the Arena itself. We would like to further emphasize that K-Sort Arena provides a **robust and general characterization of human preference**, supported by the following factors:
> - **Robust probabilistic modeling**:
> K-Sort Arena adopts probabilistic modeling and Bayesian inference, which have been shown to be robust to preference noise. With sufficiently many high-quality votes, the resulting leaderboard is highly reliable.
> - **High-quality crowd voting**:
> All crowdsourced evaluators are professors or graduate students specializing in visual generation. They also undergo pre-voting training, particularly on the evaluation criteria, as detailed in Appendix A. This significantly reduces voting noise and ensures that the collected preferences faithfully reflect human judgment.
> - **Large-scale community participation**:
> Thanks to the open-source community, K-Sort Arena has accumulated over 3,000 user votes. According to Hugging Face analytics, the project has received more than 10,000 meaningful user interactions (not just page visits). This ensures that the Arena benefits from a substantial amount of high-quality preference data.
>
> Thus, the robustness of the algorithm, the quality of the votes, and the volume of the collected data collectively ensure the reliability of K-Sort Arena, and thereby the effectiveness of K-Sort Eval.

---

> > ### Author Response · Authors · 2025-11-19
> >
> > > W4: Limited Ablation on VLM Choice: The choice of VLM (GPT-4o, Qwen-VL-Max) is critical, yet its impact is not deeply explored. The ablation study (Table 5) shows that removing posterior correction dramatically changes the model rankings, proving the VLM's raw judgment is unreliable. This makes the system highly dependent on the (unexplored) interaction between the specific VLM and the specific correction dataset.
> > >
> >
> > Thanks for your nice comment.
> > As you pointed out, directly relying on a VLM's raw judgments can indeed be unreliable, which further **highlights the importance of our proposed posterior correction**. To better illustrate its effect on stabilizing VLM choices, we conducted experiments on the FLUX.1-dev model using different VLM judges. The results are shown below:
> >
> > |                | K-Sort Arena | GPT-4o | Qwen-VL-Max | Gemini 2.5 flash | DeepSeek VL2 |
> > |----------------|--------------|--------|-------------|------------------|--------------|
> > | **Rank**       | 5            | 5      | 5           | 5                | 5            |
> > | **Score**      | 28.83        | 28.86  | 28.80       | 28.85            | 28.81        |
> > | **# Runs**     | –            | 81     | 84          | 84               | 85           |
> >
> >
> > From these results, we observe that across different VLM judges, the posterior correction consistently yields accurate ranks and scores. The only variation appears in the number of runs: judges with lower correlation produce smaller update steps, thus requiring more iterations. This demonstrates that K-Sort Eval, together with the posterior correction, is **robust to the choice of VLM judges**.
> >
> >
> >
> >
> > > Q2: The dataset curation (Section 3.1) uses a correlation threshold to filter out human votes that deviate from the global leaderboard. How did you ensure that this process doesn't just "bake in" the specific biases of the K-Sort Arena leaderboard and remove genuinely diverse or high-variance (but still valid) human preferences?
> > >
> >
> > Thanks for your thoughtful comment.
> > As discussed in Appendix B, we analyze the choice of the correlation threshold to **balance both validity and diversity**. To further address your concern, we provide a more detailed clarification:
> > - As explained in our response to W2, the robustness of the algorithm, the quality of the votes, and the volume of collected data together ensure the reliability of the K-Sort Arena leaderboard. Therefore, the correlation threshold is used primarily to **filter out clearly unreliable votes**, such as random clicking, accidental selections, or self-contradictory orderings, so as to preserve evaluation validity.
> > - The threshold of 0.75 does **not force human preferences to fully align** with the leaderboard. It still allows substantial disagreement, including ties between two models, inversions, and natural high-variance human preferences. The only samples removed are highly noisy ones exhibiting cyclic inconsistencies across three or more models. This approach maintains sample diversity while ensuring validity, and it does not induce overfitting to the leaderboard.

---

> > > ### Author Response · Authors · 2025-11-19
> > >
> > > > Q3: The ablation study (Table 5) shows very large rank changes when posterior correction is removed. This suggests the VLM's "raw" judgment is significantly different from the human-aligned one. Could you provide more analysis on the nature of this discrepancy? For example, is the VLM consistently biased on certain criteria (e.g., aesthetics vs. alignment) or for certain models?
> > > >
> > >
> > > Thanks for your nice question.
> > > Following your suggestion, we conducted two additional experiments to further analyze the characteristics of VLM bias:
> > >
> > > (1) For a single model (FLUX.1-dev), we measured the observation error at different stages of the evaluation process. Specifically, we computed the average Spearman's rank correlation coefficient within four intervals of runs: [0, 20), [20, 40), [40, 60), [60, 81). The results are shown below:
> > >
> > > | Interval   | [0, 20) | [20, 40) | [40, 60) | [60, 81) |
> > > |------------|---------|----------|----------|----------|
> > > | Avg Correlation coef. | 0.63    | 0.64     | 0.60     | 0.62     |
> > >
> > >
> > > (2) For three models with different capability levels, we measured the observation error by computing the average Spearman's rank correlation coefficient at all evaluation runs. The results are as follows:
> > >
> > > | Model                     | FLUX.1-dev (rank 5) | Realvisxl-v3.0 (rank 16) | SD-v1.5 (rank 29) |
> > > |---------------------------|----------------------|---------------------------|--------------------|
> > > | Avg Correlation coef.     | 0.62                 | 0.64                      | 0.61               |
> > >
> > >
> > > Based on these results, we observe that the discrepancy between VLM judgments and true human preferences **remains relatively stable**: both across different stages of the evaluation process and across models with varying capability levels. Therefore, this discrepancy can be viewed as "uniform", rather than being biased toward certain criteria (e.g., aesthetics vs. alignment) or for certain models.
> > >
> > >
> > > In addition, we would like to further clarify that, as noted in line 347 of the paper, we employ both swapping operations and rule augmentation, which can effectively **reduce potential VLM biases**.
> > > The swapping operation randomly permutes the image order before the VLM performs ranking, thereby mitigating position-related biases.
> > > The rule augmentation step provides the VLM with sufficiently explicit and detailed prompts, illustrated in Appendix D, to guide the judgment process. In particular, we specify clear scoring criteria across different dimensions (e.g., photorealism (30%), entity matching (30%)), which helps effectively mitigate VLM biases toward certain criteria (e.g., aesthetics vs. alignment).
> > >
> > >
> > > > Q4: How sensitive is the dynamic matching strategy (Eq. 17) to the hyperparameters, particularly the balancing coefficient? The paper states it was set after a simple grid search, but it seems crucial for balancing the exploration of diverse model matchups versus challenging the new model against its closest competitors.
> > > >
> > >
> > > Thanks for your great comment. Due to space limitations in the main paper, the grid-search results for the hyperparameter α were placed in Appendix F. We apologize for any confusion this may have caused. For your convenience, we present the search results for α in {0.1, 0.3, 0.5, 0.7, 0.9} below:
> > >
> > > | α              | K-Sort Arena | 0.1  | 0.3  | 0.5      | 0.7  | 0.9  |
> > > |----------------|--------------|------|------|----------|------|------|
> > > | **Rank**       | 6            | 6    | 6    | **6**    | 6    | 5    |
> > > | **Score**      | 28.82        | 28.77| 28.80| **28.86**| 28.72| 28.93|
> > > | **# Runs**     | –            | 107  | 89   | **81**   | 84   | 90   |
> > >
> > >
> > >
> > > Based on these results, setting α = 0.5 achieves **the best overall trade-off**.

---

### Official Review · Reviewer_2oDh · 2025-11-01

**Soundness:** 3
**Presentation:** 3
**Contribution:** 3
**Rating:** 4
**Confidence:** 4

**Summary:**

This paper introduces K-Sort Eval, an efficient framework for evaluating visual generative models. The method addresses the limitations of existing approaches: human evaluation is costly and time-consuming, while using Vision-Language Models (VLMs) as judges suffers from inherent bias and hallucinations .

**Strengths:**

1. Reliable VLM Correction: The core strength is the posterior correction method. By modeling the VLM-human preference gap as observation noise and correcting the Bayesian update , the framework significantly improves alignment with human judgments .
2. Strong Empirical Validation: The method demonstrates high consistency with the human-voted K-Sort Arena leaderboard on both image and video tasks. Its utility is also shown in practical use cases, like evaluating compressed models .

**Weaknesses:**

1. Simplified Noise Model: Assumption 1—that VLM noise is statistically independent of the true model capability—is a strong simplification. It's plausible that VLM biases are systematic (e.g., favoring certain aesthetics or failing to spot specific artifacts), which this noise model would not capture.
2. Incomplete Literature Review: The "Large Model as a Judge" related work section overlooks several recent and relevant works exploring VLMs as human-aligned evaluators. For instance, it misses applications in other domains like text-to-3D generation (e.g., **"GPT-4V(ision) is a Human-Aligned Evaluator for Text-to-3D Generation"**) or specialized tasks like personalized image generation (e.g., "DreamBench++"), which could offer broader context.

**Questions:**

1. Have you tested the framework's robustness using different VLM judges? How much does the correction mechanism's effectiveness depend on using a top-tier model like GPT-4o?
2. Can you provide more justification for Assumption 1? Did you test for systematic VLM biases (e.g., errors correlating with model rank), which would violate this assumption?

---

> ### Author Response · Authors · 2025-11-19
>
> Thank you for your careful review and constructive insights. We provide detailed replies below and hope they clarify the points you raised.
>
> ---
>
> > W1&Q2: Simplified Noise Model: Assumption 1, VLM noise is statistically independent of the true model capability, is a strong simplification. Can you provide more justification for Assumption 1? Did you test for systematic VLM biases (e.g., errors correlating with model rank), which would violate this assumption?
> >
>
> Thanks for your thoughtful comment.
> To support Assumption 1, we conducted two sets of experiments to verify its validity.
>
> (1) For a single model (FLUX.1-dev), we measured the observation error at different stages of the evaluation process. Specifically, we computed the average Spearman's rank correlation coefficient within four intervals of runs: [0, 20), [20, 40), [40, 60), [60, 81). The results are shown below:
>
> | Interval   | [0, 20) | [20, 40) | [40, 60) | [60, 81) |
> |------------|---------|----------|----------|----------|
> | Avg Correlation coef. | 0.63    | 0.64     | 0.60     | 0.62     |
>
>
> (2) For three models with different capability levels, we measured the observation error by computing the average Spearman's rank correlation coefficient at all evaluation runs. The results are as follows:
>
> | Model                     | FLUX.1-dev (rank 5) | Realvisxl-v3.0 (rank 16) | SD-v1.5 (rank 29) |
> |---------------------------|----------------------|---------------------------|--------------------|
> | Avg Correlation coef.     | 0.62                 | 0.64                      | 0.61               |
>
>
> Taken together, these experiments **verify the soundness of Assumption 1** and demonstrate that VLM noise is statistically independent of the true model capability.
>
> In addition, we would like to further clarify that, as noted in line 347 of the paper, we employ both swapping operations and rule augmentation, which can effectively **reduce potential VLM biases**.
> The swapping operation randomly permutes the image order before the VLM performs ranking, thereby mitigating position-related biases.
> The rule augmentation step provides the VLM with sufficiently explicit and detailed prompts, illustrated in Appendix D, to guide the judgment process. In particular, we specify clear scoring criteria across different dimensions (e.g., photorealism (30%), absence of artifacts (10%), etc.), which helps counteract common VLM biases (e.g., favoring certain aesthetics or failing to spot specific artifacts).
>
> > W2: Incomplete Literature Review: The "Large Model as a Judge" related work section overlooks several recent and relevant works exploring VLMs as human-aligned evaluators. For instance, it misses applications in other domains like text-to-3D generation or specialized tasks like personalized image generation, which could offer broader context.
> >
>
> Thanks for your great comment.
> Following your suggestion, we have added discussions of the relevant literature on VLMs as human-aligned evaluators [1][2] in the Related Work section.
> [1] Wu, Tong, et al. "Gpt-4v (ision) is a human-aligned evaluator for text-to-3d generation." Proceedings of the IEEE/CVF conference on computer vision and pattern recognition. 2024.
> [2] Peng, Yuang, et al. "DreamBench++: A Human-Aligned Benchmark for Personalized Image Generation." The Thirteenth International Conference on Learning Representations. 2025.
>
> > Q1. Have you tested the framework's robustness using different VLM judges? How much does the correction mechanism's effectiveness depend on using a top-tier model like GPT-4o?
> >
>
> Thanks for your nice comment.
> We conduct experiments on the FLUX.1-dev model using different VLM judges, and the results are shown below.
>
> |                | K-Sort Arena | GPT-4o | Qwen-VL-Max | Gemini 2.5 flash | DeepSeek VL2 |
> |----------------|--------------|--------|-------------|------------------|--------------|
> | Rank           | 5            | 5      | 5           | 5                | 5            |
> | Score          | 28.83        | 28.86  | 28.80       | 28.85            | 28.81        |
> | # Runs         | –            | 81     | 84          | 84               | 85           |
>
>
> From these results, we observe that across different VLM judges, our proposed Posterior Correction algorithm **consistently produces accurate ranks and scores**. The only noticeable variation lies in the number of runs: judges with lower correlation take slightly more iterations because they yield smaller update steps.
> Overall, these findings demonstrate that K-Sort Eval is **robust to the choice of VLM judges**.

---

### Official Review · Reviewer_Jr9d · 2025-11-01

**Soundness:** 3
**Presentation:** 3
**Contribution:** 3
**Rating:** 8
**Confidence:** 5

**Summary:**

This paper proposed K-Sort Eval, a novel framework for efficiently evaluating visual generative models by combining VLM judgments with statistical correction mechanisms informed by large-scale human preference rankings. The system dynamically selects the most informative model comparisons and applies Bayesian posterior correction to align VLM-based scores with true human preferences. This approach dramatically reduces evaluation cost and time compared to human-only leaderboards, while maintaining high agreement with actual human rankings.

**Strengths:**

S1) This paper tickles the pain-point that Its difficult to evaluating visual generative models both reliably and efficiently. Human annotations are highly accurate but expensive and infeasible to scale for every model update or new dataset. Meanwhile, fast automatic scoring using VLMs is scalable, but suffers from biases. K-Sort Eval aims to deliver a solution that preserves human-grade reliability while radically improving cost-efficiency and scalability by statistically correcting VLM judgments using robust human data.​ This is a highly original work and can be generalized to apply on other Arena works.

S2) It proposed a rigorous data cleaning using Spearman correlation filtering and prompt safety screening to ensure high data quality and annotation reliability.

S3) The experiments are well designed and showing high reliably of the proposed method.

**Weaknesses:**

W1) When this is applied in cases where there are not enough votings, the rankings might not be fully reflecting the performances. Will such behavior propagate to the new models? Also would it be possible to find an estimate on how much collected human/ VLM annotated data are enough to make sure that K-sort eval will works effectively?

**Questions:**

See weaknesses

---

> ### Author Response · Authors · 2025-11-19
>
> Thank you very much for your positive feedback of our work and for the constructive comments. We provide detailed responses to your questions below.
>
> ---
>
> > W1-1: When this is applied in cases where there are not enough votings, the rankings might not be fully reflecting the performances. Will such behavior propagate to the new models?
> >
>
> Thanks for your great comment.
> When the amount of voting data is insufficient, the rankings in the Arena leaderboard may not be fully reliable, which could potentially affect the evaluation of new models in K-Sort Eval.
> Therefore, K-Sort Eval uses the **official Arena leaderboard** that we provide. This ensures both the reliability of the leaderboard and the reproducibility and fairness of the evaluation process.
>
> In addition, as the Arena continuously receives new votes and incorporates new models, we promise to **periodically release updated versions** of K-Sort Eval (V1, V2, V3, ...) while updating the Arena leaderboard. Each version will include the corresponding Arena leaderboard as well as the curated dataset used for evaluation. This ensures that K-Sort Eval remains up-to-date and aligned with the evolving Arena data.
>
>
> > W1-2: Also would it be possible to find an estimate on how much collected human/ VLM annotated data are enough to make sure that K-sort eval will works effectively?
> >
>
> Thanks for your thoughtful question.
> Whether we are checking the convergence of the Arena leaderboard or determining when the evaluation of a new model in K-Sort Eval should stop, our probabilistic modeling provides **a natural and intuitive criterion**. In this model, each model's ability is represented as
> (μ,σ), where μ denotes the estimated ability and σ reflects the uncertainty of that estimate. As more votes are collected, μ converges toward the true value, and σ decreases accordingly. Therefore, evaluating the magnitude of σ directly allows us to determine whether sufficient voting information has been obtained.
> In our paper, we set the threshold for σ to 0.75; once σ<0.75, we consider the estimated ability score to be sufficiently reliable.

---

> > ### Comment · Reviewer_Jr9d · 2025-11-24
> >
> > Thanks for the rebuttal, the solutions make sense to me. I will keep the positive rating.

---

> > > ### Author Response · Authors · 2025-11-25
> > >
> > > Thank you for your encouraging response. We truly appreciate your recognition of our work once again.

---

### Official Review · Reviewer_mWxy · 2025-11-01

**Soundness:** 2
**Presentation:** 2
**Contribution:** 3
**Rating:** 4
**Confidence:** 3

**Summary:**

This paper proposes K-Sort Eval, an automated evaluation framework for visual generative models that uses vision-language models (VLMs) as judges to approximate human preferences. The method builds upon K-Sort Arena by curating a high-quality dataset from human votes and introducing two key innovations: (1) posterior correction using Bayesian updating that accounts for VLM-human alignment consistency, and (2) dynamic matching strategy that selects informative instances based on uncertainty and diversity criteria. The framework aims to provide scalable, efficient, and reliable evaluation while maintaining alignment with human preferences from K-Sort Arena.

**Strengths:**

1. The paper addresses a real scalability challenge with Arena-style human evaluations, which are costly and time-consuming. The proposed VLM-as-judge approach is timely and practical.
2. The posterior correction method (Section 3.2) is theoretically grounded in Bayesian inference, with formal derivations provided. The treatment of VLM-human misalignment as observation noise is elegant and well-justified through Lemma 1.
3. The paper demonstrates consistency with K-Sort Arena across multiple models while requiring significantly fewer evaluations (<90 runs vs. traversing entire datasets). This is quite helpful for people who wants to quickly evaluate a new model without access of huge amounts of human annotation resources.

**Weaknesses:**

1. The K-sort Eval still an existing K-Sort Arena data as supervision, which limits where this can be applied and cannot produce a score independently. Therefore, the quality of the evaluation also depends on the quality of K-sort arena data.
2. The evaluation setting more clear about how they define the evaluation task. The K-sort eval selects a model already in the K-sort arena and then predicts its rank using exisitng information from the K-sort arena, there might be information leaking during the prediction that affects the evaluation, thus making the evaluation unfair. This authors should be more clear about this.

**Questions:**

1. Can you be more clear about how the evlauation is set up? You select a model from the K-sort arena leaderboard and predicts its rank, using the data in the K-sort arena. But this model is already been compared with multiple models in the arena, will this cause circular information leakink?
2. In Table 3, why there are only 5 models for the text-to-image and text-to-video tasks, while the total rank can be as big as 12. I think you should run K-eval for all of the models in the leaderboard and summarize the prediction result, instead of picking some of them for presentation?

---

> ### Author Response · Authors · 2025-11-19
>
> We sincerely appreciate your review and the insightful comments on our work. We provide detailed responses below and hope they address your concerns.
>
> ---
>
> > W1: The K-sort Eval still an existing K-Sort Arena data as supervision, which limits where this can be applied and cannot produce a score independently. Therefore, the quality of the evaluation also depends on the quality of K-sort arena data.
> >
>
> Thanks for your great comment.
> To correct the evaluation results of VLMs, we use the Arena data as supervision. As you pointed out, relying on Arena data may raise concerns about "independence". However, this does not limit the effectiveness or generalizability of K-Sort Eval. We clarify this as follows:
> - The evaluation quality does depend on the quality of the K-Sort Arena data, but it is important to emphasize that such **quality is well guaranteed** through both the leaderboard-construction algorithm in K-Sort Arena and the data-filtering mechanism in K-Sort Eval. Specifically, K-Sort Arena adopts probabilistic modeling and Bayesian inference, and it has accumulated a sufficiently large number of human votes (over 15,000 pairwise comparisons), ensuring the reliability of the constructed leaderboard. In addition, K-Sort Eval uses Spearman's rank correlation coefficient to assess consistency with the leaderboard and perform data filtering and curation. Consequently, the resulting dataset is of high quality, which in turn ensures the reliability of K-Sort Eval.
> - Based on the K-Sort Arena data, the application scope of K-Sort Eval is broad and unrestricted. In our experiments, we chose to evaluate the models included in the Arena primarily for the sake of intuitive comparison. In practice, K-Sort Eval can **evaluate any new model**, especially models that have never appeared in the Arena before. Therefore, although K-Sort Eval relies on Arena data, this dependence does not limit its applicability.
>
> > W2&Q1: Can you be more clear about how the evlauation is set up? You select a model from the K-sort arena leaderboard and predicts its rank, using the data in the K-sort arena. But this model is already been compared with multiple models in the arena, will this cause circular information leakink?
> >
>
> Thanks for your thoughtful question.
> We provide a more detailed explanation of the evaluation pipeline and setup. For any given model, the procedure is as follows:
> (1) Dynamic instance matching (Eq. 17): Using Eq. 17, we dynamically match a set of instances from the dataset. Each instance contains the outputs of K models (A, B, C, D) along with the human-annotated ranking (e.g., [A>C>B>D]).
> (2) Augmenting with the new model:
> The output of the new model, denoted as E, is added to the K existing outputs, forming a group of K+1 images.
> (3) Ranking by the VLM:
> The VLM is then asked to rank the K+1 images, producing a result such as [A>B>E>C>D].
> (4) Assessing reliability using human annotations (Eq. 12):
> We compute the correlation between the VLM's ordering of the K known outputs (A, B, C, D) and the human-annotated ranking, i.e., [A>B>C>D] vs. [A>C>B>D]. This correlation serves as a measure of reliability for the current evaluation, as defined in Eq. 12.
> (5) Posterior estimation and correction (Eq. 14):
> We compute the posterior probability of the new model's ability. Then, based on the reliability score from Step 4, we apply a correction to this posterior using Eq. 14.
> (6) Iterative evaluation:
> Steps 1-5 are repeated until the uncertainty of the new model's estimated ability falls below a predefined threshold.
>
> This constitutes the full evaluation process. Importantly, although the K-Sort Arena data (i.e., the per-instance sets of K images with human rankings) is used, **no leaderboard information**, such as model ability scores or global rankings, is ever used or accessed during the VLM judgment process. The leaderboard is **only** used after all evaluations are completed, for validating the accuracy of the estimated model abilities. Therefore, throughout the evaluation, we can ensure that no leaderboard information is leaked or injected into the assessment process.

---

> > ### Author Response · Authors · 2025-11-19
> >
> > > Q2: In Table 3, why there are only 5 models for the text-to-image and text-to-video tasks, while the total rank can be as big as 12. I think you should run K-eval for all of the models in the leaderboard and summarize the prediction result, instead of picking some of them for presentation?
> > >
> >
> > Thanks for your nice comment.
> > K-Sort Eval is designed to evaluate a new model and provide two metrics: an absolute score and a relative score. The absolute score reflects the model's estimated ability, while the relative score corresponds to its ranking on the K-Sort Arena leaderboard. Here, this ranking is obtained by **indexing the Arena leaderboard** using the model's estimated ability score. Therefore, in Table 3, each model is **evaluated independently**, with the goal of assessing its own ability, rather than constructing a new leaderboard from these models.
> >
> >
> > In addition, in Table 3, we selected representative models from different regions of the K-Sort Arena leaderboard, covering both high-ability and low-ability models. This demonstrates that K-Sort Eval can reliably evaluate models with varying capability levels. As you pointed out, evaluating all models rather than a subset would indeed provide a more comprehensive and convincing comparison. In response, we have evaluated all available models using K-Sort Eval and compared the results directly with the K-Sort Arena leaderboard.
> > It is worth noting, however, that some commercial models are neither open-sourced nor accessible via API. As a result, we evaluate only the models supported by the Replicate API. Due to space constraints, we report results for the top 15 consecutive text-to-image models on the Arena leaderboard:
> >
> > | Model              | K-Sort Arena Rank | K-Sort Arena Score | K-Sort Eval Rank | K-Sort Eval Score |
> > |--------------------|-------------------|---------------------|------------------|--------------------|
> > | GPT-4o             | 1                 | 30.86               | –                | –                  |
> > | FLUX-1.1-pro       | 2                 | 29.52               | 2                | 29.59              |
> > | SD-v3.5-large      | 3                 | 28.97               | 3                | 28.97              |
> > | FLUX.1-pro         | 4                 | 28.90               | 4                | 28.92              |
> > | FLUX.1-dev         | 5                 | 28.83               | 5                | 28.86              |
> > | Aurora             | 6                 | 28.72               | –                | –                  |
> > | Midjourney-v6.0    | 7                 | 28.64               | –                | –                  |
> > | Dalle-3            | 8                 | 28.27               | –                | –                  |
> > | SD-v3.5-large-turbo| 9                 | 27.73               | 9                | 27.77              |
> > | FLUX.1-schnell     | 10                | 27.71               | 10               | 27.67              |
> > | Midjourney-v5.0    | 11                | 27.44               | –                | –                  |
> > | SD-v3.0            | 12                | 27.13               | 12               | 27.08              |
> > | Pixart-Sigma       | 13                | 26.38               | 13               | 26.39              |
> > | Proteus-v0.2       | 14                | 24.69               | 14               | 24.70              |
> > | Open-Dalle-v1.1    | 15                | 24.65               | 15               | 24.66              |

---

### Author Response · Authors · 2025-12-02
**Rebuttal Summary of Submission 15284**

We are very grateful for the thoughtful and professional comments from the reviewers. Below is a summary of our response:

First, we are delighted to see that all reviewers **explicitly recognized the novelty and practicality** of taking VLM-human misalignment as observation for posterior correction, exemplified by remarks such as "*This is a highly original work and can be generalized*" (Reviewer Jr9d), "*a novel and clever approach*" (Reviewer 9sHD), "*This is quite helpful*" (Reviewer mWxy), and "*shown in practical use cases*" (Reviewer 2oDh). Furthermore, they praised the **significance of our experiments**, for instance, "demonstrates consistency while requiring *significantly fewer evaluations*" (Reviewer 9sHD). In particular, Reviewer Jr9d gave us a *positive score of 8* and maintained this score after the rebuttal.

Second, the reviewers also provided insightful suggestions, primarily in four major aspects, which we have addressed in detail through experiments and explanations. The summary is as follows (W: Weekness, Q: Question):

- *Scalability of K-sort Eval* (mWxy W1, 9sHD W1&W3&Q1)
To correct the evaluation results of VLMs, we use the Arena data as supervision. It is important to note that while relying on Arena data, K-Sort Eval **can evaluate any new model**, especially models not in Arena. Moreover, as Arena is continuously maintained, we promise to **periodically release updated versions** of K-Sort Eval (V1, V2, V3, ...), which will include a new leaderboard and dataset for evaluation. Therefore, K-sort Eval possesses high scalability.

- *Accuracy and Fairness of Evaluation* (mWxy W2&Q1, Jr9d W1)
Firstly, the Arena data, through Bayesian modeling, crowdsourced voting, and a consistency filtering strategy, **ensures data quality**. Secondly, the evaluation process is guaranteed to **avoid circular information leaking** (the evaluation setup has been thoroughly clarified in the response to mWxy W2&Q1). Furthermore, as detailed in the response to Jr9d W1, due to the probabilistic modeling, there is a **natural criterion** to determine when the evaluation process is complete. Thus, the accuracy and fairness of the evaluation are well-guaranteed.

- *Validity of Assumption 1* (2oDh W1&Q2)
In the Rebuttal, we conducted two sets of experiments to verify its validity, including: For a single model (FLUX.1-dev), we measured the observation error at different stages of the evaluation process; and for three models with different capability levels, we measured the observation error by computing the average Spearman's rank correlation coefficient at all evaluation runs. The results demonstrate that VLM noise is **statistically independent** of the true model capability, thereby verifying the soundness of Assumption 1.

- *Robustness to Different VLM Judges* (2oDh Q1, 9sHD W4)
In the Rebuttal, we conducted experiments on the FLUX.1-dev model using different VLM judges. We observe that across different VLM judges, including GPT-4o, Qwen-VL-Max, Gemini 2.5 flash, and DeepSeek VL2, the posterior correction **consistently yields accurate** ranks and scores. Therefore, the proposed posterior correction is robust to the choice of VLM judges.

Although we did not have chance for extensive discussion with the reviewers, we believe the experiments and explanations in the Rebuttal can address the reviewers' concerns. Finally, we thank the reviewers once again for their comments, and especially thank the ACs for their time and effort.

---

### Meta-Review · Area_Chair_a1dj · 2026-01-07

**Summary:**

The paper addresses the need for scalable, human-aligned evaluation of visual generative models. Crowdsourced platforms like Arena provide reliable human preferences but are costly and slow. To improve efficiency, the authors propose K-Sort Eval, a VLM-based evaluation framework that combines posterior correction and dynamic matching. Using a dataset curated from thousands of human votes, K-Sort Eval performs (K+1)-wise comparisons between models, corrects VLM predictions based on human consistency, and selects comparisons to maximize information gain.

The reviewers highlight that the proposed evaluation tackles the scalability challenge of Arena-style human evaluations, which are costly and slow, by using VLMs as judges. The proposed posterior correction method is theoretically grounded in Bayesian inference, treating VLM-human misalignment as observation noise. Experiments show that the method produces results consistent with K-Sort Arena while requiring far fewer evaluations (<90 runs), enabling efficient assessment of new models without extensive human annotation.

The authors addressed the reviews and concerns. I recommend for acceptance.

**Reviewer Concerns:**

The authors addressed all reviews.

**Reviewer Scores:**

Given that the reviewers did not have the opportunity to respond to the rebuttal, it is plausible that they may revise their initial scores upon further consideration.

---

### Decision · Program_Chairs · 2026-01-26

Accept (Poster)